# Ceramic Materials for Biomedical Applications: An Overview on Properties and Fabrication Processes

**DOI:** 10.3390/jfb14030146

**Published:** 2023-03-04

**Authors:** Lorenzo Vaiani, Antonio Boccaccio, Antonio Emmanuele Uva, Gianfranco Palumbo, Antonio Piccininni, Pasquale Guglielmi, Stefania Cantore, Luigi Santacroce, Ioannis Alexandros Charitos, Andrea Ballini

**Affiliations:** 1Department of Mechanics, Mathematics and Management, Polytechnic University of Bari, Via Orabona 4, 70125 Bari, Italy; 2Independent Researcher, Sorriso & Benessere-Ricerca e Clinica, 70129 Bari, Italy; 3Microbiology and Virology Unit, Department of Interdisciplinary Medicine, University of Bari “Aldo Moro”, 70126 Bari, Italy; 4Emergency/Urgency Department, National Poisoning Center, Riuniti University Hospital of Foggia, 71122 Foggia, Italy; 5Department of Precision Medicine, University of Campania “Luigi Vanvitelli”, 80138 Naples, Italy

**Keywords:** biomaterials, bioceramics, dentistry, bone tissue engineering

## Abstract

A growing interest in creating advanced biomaterials with specific physical and chemical properties is currently being observed. These high-standard materials must be capable to integrate into biological environments such as the oral cavity or other anatomical regions in the human body. Given these requirements, ceramic biomaterials offer a feasible solution in terms of mechanical strength, biological functionality, and biocompatibility. In this review, the fundamental physical, chemical, and mechanical properties of the main ceramic biomaterials and ceramic nanocomposites are drawn, along with some primary related applications in biomedical fields, such as orthopedics, dentistry, and regenerative medicine. Furthermore, an in-depth focus on bone-tissue engineering and biomimetic ceramic scaffold design and fabrication is presented.

## 1. Introduction

The word “biomaterial” refers to a substance or a mix of materials of synthetic or natural origin interacting with biological systems [1]. The main purpose of biomaterials is to support the healing or replacement of an organ in a human body that has been altered by a disease or an accidental event and to successfully restore function and sometimes aesthetic features without endangering human life [2]. Biomaterials can be classified according to their chemical nature as metallic, polymeric, ceramic, and composite, and can also be biologically derived [3]. Care must be taken when defining a biomaterial as biocompatible; in fact, biocompatibility is an application-specific property: a biomaterial that is biocompatible or appropriate for one application may not be biocompatible for another one [4]. Among implantable biomaterials, science and technology are disclosing new possibilities through combinations with new materials, new coatings, new design approaches, and new manufacturing technologies (biomimetic and functional materials, nanotechnologies, tissue engineering, computational methods for simulation, additive manufacturing, and many others) that will revolutionize the field of biomaterials in the short term [5,6].

The application of a specific biomaterial is driven by the necessary composition, material properties, structure, and triggering of desired in vivo reactions in order to perform a precise function. Furthermore, for usage in the medical field, researchers need to pay attention to bioethics, biocompatibility, bioabsorbency, and toxicity [7,8]. A possible categorization of different specific properties of a biomaterial in order to be employed in medical applications in order to maximize functional results is presented in Figure 1. As an example, in tissue-engineering applications, biomaterials must be capable of being modeled to the proper shape and size of the section of the organic part to be replaced, and the surface of the replacement part must possess a precise roughness to inducing cell adhesion and to favor biological integration with the tissues or the skeleton, whereas the inner topology of the replacement part must present a porous bulk, as described in the following sections.

Compared to other biomaterials such as metals or polymers, bioceramics possess a unique combination of properties, such as (i) high intrinsic strength—materials such as alumina and zirconia show great mechanical properties such as high wear resistance and low coefficient of friction, making them suitable for usage in high-stress applications such as artificial joints or dental implants; (ii) biocompatibility—bioceramics are, in general, compatible with human tissues, reducing the risk of adverse reactions or inflammation, and some bioceramics in particular, such as hydroxyapatite or bioactive glasses, show bioactive behaviors that can promote tissue regeneration and osteointegration; (iii) versatility—bioceramics can be modeled to precise shapes and their compositions can be tailored to enhance specific properties. All of these features make bioceramics an adequate solution for a wide variety of biomedical issues. Research on ceramic biomaterials is developing rapidly, finding new key applications in medicine and biotechnology, especially in terms of their usage as load-bearing parts, joint replacements, fillers, veneering materials, drug-delivery platforms, and biomimetic scaffolds [9].

In this article, a brief review of the fundamental physical, chemical, and mechanical properties of ceramic biomaterials is given, and the various types are described according to the international literature. The biocompatibility of the materials to a biological environment surrounding a ceramic implant is highlighted [10]. Several insights concerning primary applications in biomedical fields, such as orthopedics, dentistry, and bone-tissue engineering, are provided. In Section 5, an in-depth focus on biomimetic-scaffold design and fabrication is presented.

## 2. Properties of Bioceramic Materials

### 2.1. Physical and Chemical Properties of Ceramics

The term “ceramic” (from the Greek word κεραμικό: “keramikò,” which means “burnt stuff”), a word that is also found in ancient texts [11], indicates any heat-treated material derived from clayey raw materials through a process called firing. Generally speaking, ceramics are inorganic materials consisting of metallic and non-metallic components chemically bonded together by means of ionic or prevalently ionic bonds with a variable degree of covalent character [12,13]. Their properties essentially depend on the type of these bonds and on the type of plexus that shapes the microstructure of the respective material [3]. They can be both crystalline and non-crystalline, and the structure in which a crystalline phase is dispersed in a non-crystalline one is very common. A ceramic material can typically be identified as a member of one of these categories: glasses, structural clays, whitewares, refractories, abrasives, cements, or advanced ceramics [12]. The main characteristics of ceramic materials are high stiffness and strength, great hardness, insulating behavior, and resistance to high temperatures, wear, and chemical degradation, and in fact most bioceramics are not reactive within the living body [3,11]. Their low toughness and therefore great fragility is a major issue.

Depending on the chemical properties (molecular bioactivity when interacting with human organisms), ceramic biomaterials can be classified as (a) inert, (b) low or medium surface activity, or (c) bioresorbable (adsorbable) ceramics. The choice of the material to be used (inert, bioactive, or bioresorbable) depends on the function to be accomplished in each specific application. Inert bioceramics such as alumina (Al_2_O_3_) do not promote the connection with living tissues, can withstand low-pH environments for thousands of hours, and possess high chemical inertia, which in turn means that they require a long time until stable connections between implants and tissues are established. Once implanted, they are surrounded by a network of fibrous connective tissue of varying thickness, which holds the implant and at the same time isolates it from adjacent tissues. Therefore, due to their high biocompatibility and mechanical strength, they are designed for permanent implants [14]. Low- and medium-activity materials, in addition to binding to specific proteins, can also release ions, thus promoting the integration of implants to living tissues [15]. Finally, bioabsorbable ceramics are destined to remain until the regeneration of the new tissue where they are inserted occurs.

### 2.2. Mechanical Properties of Ceramics

Regarding the essential mechanical properties of ceramics, these include (i) low resistance to tensile loads, (ii) high hardness due to the internal microstructure characterized by strong ionic or covalent bonds, and (iii) minor or negligible plasticity resulting in low fracture toughness and resistance to shock loads. Different from metal alloys, which exhibit a ductile behavior [16], the main disadvantage of ceramic materials is their susceptibility to brittle fracture, even in the presence of very low energy absorption. Even if they are brittle, ceramics have greater hardness and elasticity values than metals. At room temperature, in both crystalline and non-crystalline ceramics, when undergoing tensile stresses fractures are observed before any plastic deformation occurs, and this effect also leads to poor fatigue resistance. The brittleness consists of the formation and propagation of cracks in a direction perpendicular to the applied load. In crystalline ceramics, cracks develop both transgranularly, i.e., through grains, and along certain crystallographic planes presenting high atomic density. These weak responses occur because at micro-scales a wide variety of imperfections of different size and geometry exists, such as internal pores, micro-cracks, grain misalignments, impurities, microscopic notches, and so on [17]. These defects are usually formed in the production process and are generally caused by thermal gradients induced by heat cycles. In general, ceramics with finer microstructures show better mechanical properties.

Since ceramics have various imperfections, it is important to know what the maximum allowable stress on a component can be. The ability of a material to resist the destructive propagation of a crack is called fracture toughness, and the related metric is the K_IC_ coefficient, which is defined as the value of the stress for a pre-existing crack in a standard specimen for propagating it rapidly by pulling the specimen. For ceramics it is quite low compared to metals (typical values are lower than 10 MPa·m^1/2^, usually in the range of 2–4 MPa·m^1/2^) [18]. The resistance of ceramics to tensile loads is much lower than their resistance to compressive loads. Whereas for metals the maximum allowable stress measured in compression is equal to that of tension, in the case of brittle materials it is about 15 times greater [19,20]. Furthermore, it is important to know that, for ceramics, fracture toughness depends not only on the actual tensile stress but also on the duration of its application [21,22].

Strength attenuation and the ultimate failure of ceramics can occur without cyclic loading, and the mechanism controlling material failure is chemical rather than mechanical, but in general, the kinetics of the chemical decomposition of ceramics is characterized by very slow rates. They also show a very high specific elasticity value (E/ρ), and therefore, ceramic fibers are used as a reinforcement for composite materials. Furthermore, it has been observed that the elasticity (E) for ceramics remains almost constant at high temperatures, allowing their use in components operating in hot environments [23].

## 3. Ceramic Materials for Biomedical Applications

### 3.1. Bioceramics for General Applications

The science of ceramics is developing rapidly, as ceramics can be porous or glassy and hence can have many applications in medicine and biotechnology. They are widely used in dental and orthopedic applications for wound healing and tissue engineering when non-metallic inorganic materials are required. Bioceramics can be designed to mimic the mechanical properties of the surrounding tissues, and this can improve the long-term stability of the implant. For biomedical applications, these materials can also be used for the fabrication of all-ceramic prosthetic components and can be distinguished according to their glass-content structure in (i) mainly glass, (ii) glass mass filled with other particles, and (iii) polycrystalline [3,11,24]. These biomaterials can be crystalline (sapphire), polycrystalline (alumina, hydroxyapatite), glass–ceramic (Ceravital), and composite. As described in the following sections, bioactive and bioresorbable ceramic materials are currently employed to repair and reconstruct diseased or damaged parts of the musculoskeletal system by inserting customized supporting structures called biomimetic scaffolds” in the fracture site [11,25]. Obviously, the choice of the correct bioceramic depends on the site of application.

Alumina (Al_2_O_3_) and zirconia (ZrO_2_) are the two most important ceramic oxides for biomedical purposes, which are used for damaged bone tissue and joint repair and replacement, as in the case of total-hip and -knee arthroplasty, due to their excellent wear resistance and biocompatibility. They are inert materials but can be used in combination with other materials, such as biodegradable polymers, to deliver drugs and promote tissue regeneration. The biocompatibility of these materials is related to the chemical stability of the crystal lattice, a symmetrical three-dimensional structural arrangement of the constituent ions inside a crystalline solid, which gives alumina and zirconia high anticorrosive performance and reliable in vivo behavior. Free hydroxyl radicals (-OH) are commonly found on the surfaces of implants realized with these materials, which interact with body fluids, providing a lubricating layer around the implants. The mechanical strength, fatigue strength, and brittleness of Al_2_O_3_ depend on the purity, size, and distribution of its crystals, as well as its density. Due to the high mechanical strength, it is used to produce endosseous implants both in orthopedics and in maxillofacial surgery [26]. Implants made of Al_2_O_3_ combine small average grain size (<4 μm) and low surface roughness (Ra ≤ 0.02 μm), showing excellent tribological properties [27]. Pure zirconia has a single crystal structure at room temperature and transitions to a tetragonal and cubic structure at higher temperatures. To stabilize the grid of the square and the cubic structure of zirconia, various oxides are added, among which are magnesium oxide, yttrium oxide, calcium oxide, and cerium oxide (Ce_2_O_3_) [28]. Pure zirconia occurs in three main crystalline-phase structures: cubic (c), tetragonal (t), and monoclinic (m). Microcracks in the crystal-mesh structure of zirconia are self-limiting if the transformation from a tetragonal to a monoclinic crystal structure is controlled. Zirconium oxide exhibits more than twice the strength of polycrystalline aluminum oxide, a lower modulus of elasticity, and greater brittleness [29]. A huge number of zirconia femoral ball heads have been implanted with good results in terms of biocompatibility and mechanical behavior, as well as immovable prosthetic works and dental implants [30]. By adding CaO, MgO, and Y_2_O_3_ oxides, which stabilize the zirconia lattice, it is possible to control the transformation of the phases, thus obtaining multiphase materials such as the stabilized zirconia. The addition of 2–3% mole yttrium oxide (Y_2_O_3_) produces a partially stabilized zirconia consisting of fine square zirconia crystals. During the propagation of a crack in the mass of this material around the crack tip, a transformation of the crystals from a cubic to a monoclinic crystal system takes place [31].

Another important class of bioceramics includes calcium orthophosphates, such as hydroxyapatite (HA, Ca_10_(P0_4_)_6_(OH)_2_) and tricalcium phosphate (TCP, Ca_3_(PO_4_)_2_). In general, apatites are inorganic compounds with the general formula Ca_5_(PO_4_)_3_X_2_, where X can be fluorine ions (such as fluorapatites (FAp)), chloride ions (chloroapatites (ClAp)), or hydroxyl ions (hydroxyapatites (OHAp)). Ca_10_(PO_4_)_6_(OH)_2_ hydroxyapatite is the main structural mineral component of bones and teeth and has typically low crystallinity. The stoichiometric composition consists of 39.68% Ca and 18.45% P. As the Ca/P ratio increases, the resistance increases, reaching a maximum value for a ratio of ~1.67, and after this value it decreases [32,33]. The bone substance, although similar to hydroxyapatite, contains sodium, chlorine, and magnesium, plus other additional ionic units, and is stable at pH 9–12. The key property of hydroxyapatite lays in its high biocompatibility, thus promoting osseointegration and making it one of the most suitable materials for bone-repair and -replacement applications. For the same reasons, hydroxyapatite is commonly selected as an optimal material for dental implants. Furthermore, the advantage of this material is its ability to incorporate different chemicals and gradually attribute them to their microenvironment. It can be used as a drug-delivery platform to release therapeutic agents over a controlled period of time. In addition, it promotes bone synthesis and the regeneration of bone tissue, but its major drawback is the relatively low mechanical strength [34]. The strength decreases exponentially with increasing porosity. The Weibull’s modulus belongs to the value range of 5–18, which indicates that hydroxyapatite behaves like a typical brittle ceramic, and the Young’s modulus ranges between 35–120 MPa. The low strengths combined with the susceptibility to slow crack growth (especially in wet conditions) confirm the low-load reliability of dense hydroxyapatite implants [35]. As porosity increases, fracture toughness decreases dramatically. Noteworthy is the fact that porous hydroxyapatite ceramics are less fatigue resistant than dense hydroxyapatite. Mechanical properties can be modified by changing the percentage content of the components or the grain size of the solid phase [36]. Porous hydroxyapatite ceramics have been widely used as bone substitutes, as the porous hydroxyapatite allows contact with the bone and the pores provide a stable matrix for cell attachment and osteogenic factors. Osseous tissue develops within the pores, increasing the strength of the implant. The usual preparation method of porous hydroxyapatite ceramics (pore sizes of 100–600 µm) is through the powder-sintering process with suitable additives such as paraffin, naphthalene, and hydrogen peroxide, which allow pores to form through the gases they release at elevated temperatures [37,38]. Tricalcium phosphate has several uses, such as in maxillofacial surgery, otolaryngology, orthopedic prosthesis, neurosurgery (spinal-cord surgery), dental implants, percutaneous appliances, periodontal therapy, and alveolar incrementations [39,40]. The biochemical behavior of calcium phosphates interacting with body fluids depends on temperature and pH variations. In fact, the non-hydrated phases of calcium phosphate in a high-temperature environment interact with body fluids at 37 °C to form hydroxyapatite, which are outlined on the exposed surfaces of tricalcium phosphate [41,42]. However, calcium-phosphate cements also have some disadvantages, mainly related to poor mechanical performance, which has limited or no application in relation to pure ceramic materials [43,44].

Bioactive glasses are a unique group of synthetic bioresorbable ceramics that react in the presence of biological fluids, improving and enhancing the healing ability of human bodies. Bioactive glasses can be used in tissue engineering as scaffold materials to support tissue regeneration in several applications, including wound healing and nerve regeneration. Regarding the chemical composition, they mainly contain silica but also small quantities of some components such as Na_2_O, P_2_O_5_, and CaO. These components are very important because they determine their bioactive activity and bioabsorbability. A remarkable advantage is their mechanical strength and the possibility of being used as veneering materials [45]. Bioactive glasses are prepared either by the method of rapid cooling of a molten glass at room temperature (melt-derived glasses) to avoid crystallization or by the sol-gel method, which provides the formation of a three-dimensional porous gel network from a colloidal solution by controlling the pH value [46].

In the class of non-oxide ceramic materials, silicon nitride (Si_3_N_4_) is the most excellent, particularly for its high reliability in environments characterized by high temperatures. It has superior mechanical strength and hardness compared to alumina and is typically produced by the hot-isostatic-pressing (HIP) method [31,39]. Hardened silicon nitride, with a tensile strength of approximately 1 GPa and a stress-intensity factor of 10–12 MPa·m^1/2^, has been used in the production of femoral heads with extremely low levels of wear.

Another non-oxide ceramic is silicon carbide (SiC), which is also produced through the HIP process. This material has greater hardness and strength than alumina and a similar stress-intensity factor. The tensile strength of this material reaches 650 MPa and a stress-intensity factor of 9–10 MPa·m^1/2^. This material is particularly useful in the orthopedic field. In its preparation, the silicon-carbide bulk is covered with a layer of silicon oxide a few nanometers thick as the product of surface oxidation [40]. The main properties of the bioceramic materials presented in the current section are synthesized in Table 1.

### 3.2. Advanced Bioceramics

Over time, engineers have tried to improve ceramic materials in order to give them ultra-specialized properties through the development of composite micro- and nano-systems [57,58]. A composite material is defined as a heterogeneous combination of two or more distinct materials presenting a finite interface between them. Ceramic nanocomposites constitute an emerging research field aiming to further improve specific properties of bioceramics and to offer new opportunities for the treatment of a wide range of biomedical issues [59,60,61,62,63]. Nanocomposites are a class of composites in which one or more dimensions of the reinforcing phase is in the nanometer range (1 nm = 10 Å), typically up to 100 nm [64]. The characteristic trait of nanocomposite materials is their ability to combine properties and functionalities that are out of reach for traditional materials. By incorporating nanoparticles into a ceramic matrix (e.g., by adding organic molecules, carbon nanotubes, graphene, nanoscale ceramics, proteins, or even DNA to bioceramics or bioglasses), it is possible to create materials with improved mechanical strength, biocompatibility, and osteoconductivity [65]. Ceramic nanocomposites have been developed for a wide range of biomedical applications, including bone replacement or repair and drug delivery. In dentistry and tissue engineering, the architecture of a custom-built nanocomposite material should allow the tissues to self-organize within the organism [16]. In clinical applications, nanocomposite ceramic materials must exhibit adequate mechanical properties, including compressive strength, stiffness, fracture toughness, and fatigue resistance, as well as biocompatibility [66].

Characteristic examples of nanocomposite reinforcements include fullerenes, carbon nanotubes, layered silicates, metal nanoparticles, and dendrimers. In most cases, the necessity is to increase the bending strength, reduce the modulus of elasticity, and avoid material failure. Regarding mechanical bone reconstruction, the solid part of the bone exhibits anisotropic deformation and specific fracture resistance, characteristics that result from its complex composition (collagen fibrils and brittle hydroxyapatite-carbonate crystals). A variety of bioceramics possesses greater hardness than bone, yet several exhibit lower fracture toughness [67]. Therefore, one method of ensuring biomimetic properties occurs by developing biocompatible composites by inserting polymers such as polyethylene within a ceramic matrix of higher mechanical strength, such as sintered hydroxyapatite powder, as a second phase. A transition from ductile to brittle behavior occurs for a volume fraction of hydroxyapatite that ranges between 0.4 and 0.45. In addition, the bioceramic composite acquires a tensile strength of 22–26 MPa and increased fracture toughness for a hydroxyapatite volume fraction < 0.4 due to the deformation associated with crack propagation [68,69]. The hydroxyapatite–polyethylene composite exhibits properties close to those of bone, and when increasing the volume fraction of hydroxyapatite to 0.5 it acquires an increased modulus of elasticity of 1–8 GPa and a greatly reduced probability of failure from >90% to 3% [70]. Bioceramic nanocomposites reinforced with carbon nanotubes were shown to improve mechanical properties of ceramic matrices for scaffolds and enhance cellular proliferation and differentiation in vitro [71]. Other studies demonstrated that bioceramic nanocomposites based on hydroxyapatite matrices can be used for releasing drugs over a controlled period of time, thus promoting tissue regeneration [72]. In dentistry, nanocomposites are now being exploited as functional rather than simply structural materials through the exploitation of their biochemical properties [73,74]. Finally, the contact of dental prostheses with a biofilm from dysbiosis oral micro-biota has a strong impact on the possibility of developing periimplantitis and the infection and failure of an implant. Furthermore, even teeth with local dysbiosis adjacent to the implant could be detrimental to the survival of the implant itself [75]. The micro-surface of the implant affects the ability of the biofilm to adhere to the implant area. For this reason, scientific research must protect dental implants from this condition by developing new materials that can inhibit the adhesion of the bacterial load on the prosthesis with material that has antibacterial properties and is an inert bioceramic [76]. An optimal biomaterial would be the ceramic nanocomposite Al_2_O_3_/Ce-TZP, which has all the properties necessary for osseointegration but at the same time guarantees a reduced accumulation of bacteria near the implant in the oral cavity. In a study concerning the adhesion of standard bacteria that are very common in the oral cavity (i.e., Actinomyces naeslundii, Aggregatibacter actino-mycetemcomitans, Fusobacterium nucleatum, Porphyromonas gingivalis, Streptococcus oralis, and Veillonella parvula), several biomaterials for implants were compared: calcium hydroxyapatite, Al_2_O_3_/Ce-TZP with sterile sandblasted nanocomposites and with glassy coating, and another type of Al_2_O_3_/Ce-TZP (enriched with ZnO). The analysis showed that the adhesion of bacteria was reduced in the groups in which the coatings with antimicrobial glass materials were present. Furthermore, the one enriched with ZnO had a significant antimicrobial effect [77].

Another outstanding development in the bioceramics field is the creation of bioactive coatings. The surfaces of metal or ceramic implants can be coated with ceramic layers, bioactive molecules, or antimicrobial agents to prevent the risk of infection and promote tissue regeneration, wound healing, and osteointegration with the surrounding tissues, thus making them an effective functional material [78]. Bioceramic coatings can significantly improve the chemical stability of implants and increase osteogenic activity in vitro and in vivo. For example, a research study showed that bioactive glass coatings on zirconia implants improved osteoconductivity and biocompatibility [79]. The use of hydroxyapatite as a coating on orthopedic and dental metallic implants combines the advantages of metallic materials in terms of mechanical properties with the excellent biocompatibility and bioactivity of hydroxyapatite. In fact, this material coupling is very popular [80]. Pure metal implants do not integrate with bone and, like all bio-inert materials, are surrounded by dense fibrous tissue that prevents the desired stress distribution, with the possible result of implant loosening. However, in the case of coated implants, the bone is fully integrated with the implant even during the first functional-loading phases. Hydroxyapatite coatings perform several functions: they ensure the creation of a stable union of the implant with the bone and minimize adverse reactions of the immune system. In addition, they reduce the release of metal ions into the body and protect the metal surface from the biological environment. In the case of porous metal implants, they encourage bone growth within the pores. Finally, coating an implant with hydroxyapatite also improves its hemocompatibility [81]. During the implantation phase, there is a tendency to adhere to the platelets and a thin layer (film) of proteins is formed, which modifies the surface properties of the biomaterial. Without the addition of hydroxyapatite, this thin film is often incomplete, and when it meets blood and body fluids it leads to clots [82].

The choice of coating technique depends on the specific requirements of the application, such as the desired thickness and uniformity of the coating or the type of bioactive molecule being used. Commonly adopted coating techniques are sol-gel deposition, dip-coating, electrophoretic deposition, and plasma spraying [83,84].

Cyclic-fatigue effects, a topic well described for composite ceramic materials, represent an issue that must be considered in implant design. In most cases, it is necessary to increase the bending strength, reduce elasticity, and avoid material failure. As a positive effect, it was reported that the fracture toughness and flexural strength of bioceramics increased in wet environments [85]. To overcome the above limitation, the usage of bioceramic coatings and the development of nanocomposite ceramics should be considered as appropriate approaches [86,87].

### 3.3. Bioceramics for Dentistry Applications

The fundamental property of ceramic materials for dentistry is their compatibility with biological tissues. In recent decades, bioceramics such as alumina, zirconia, SiAlON, bioglasses, and hydroxyapatite (Ca_10_(PO_4_)_6_(OH)_2_) have been studied for dentistry applications. Porcelain, zirconium oxide, and single-crystal sapphire are already being used on a large scale for orthodontic issues [88,89,90]. The main disadvantages of modern osteoplastic devices made of bioceramics are the fragile behavior and the low resistance to tensile or bending forces. They are not osteoinductive except for bioactive glass, and bioabsorption is generally unpredictable. Indeed, TCP and synthetic HA are not bioresorbable in the short term, whereas bioactive glass is rapidly absorbed.

In many dentistry applications, a glass mixture is usually crystallized by employing alumina, zirconia, magnesium spinel (MgAl_2_O_4_), and other compounds in the forms of powders or crystals [88]. By imposing a controlled heat treatment, commonly known as ceramification or devitrification, the final result is obtained. When crystals are used, composite materials known as interpenetrating phase composites (IPC) can be formed. They are constituted by two phases (crystals and glass) that are interconnected and constantly expand inside each other without generating a chemical bond. The production of these IPCs takes place in two stages. Initially, the ceramic is sintered to form a porous core consisting of alumina- or magnesium-spinel (MgAl_2_O_4_) crystals or alumina and zirconia in a ratio of 70/30 [89]. The molten glass is then filtered through a porous mesh, and after this phase it fills all pores and gaps of a precise shape. In this way, a high-strength frame is created on which a special dental porcelain (i.e., an aesthetic coating) is deposited and fired. In the event that oxides are added in the form of powder, a specific ceramic material called glass–ceramic is formed. Commonly used reinforcing particles are mainly lithium-disilicate crystals [91,92].

Glass-matrix ceramics are based on a ternary-material system consisting of clay/kaolin, quartz (silica), and natural feldspar (a mixture of potassium and sodium aluminosilicate). Potassium feldspar (K_2_A1_2_Si_6_O_16_) forms leucite crystals (crystalline phase) that, depending on the quantity, can increase the intrinsic strength of a restoration. These bioceramic materials are used as a veneering material in metal alloys and ceramic substrates and as an aesthetic monolithic tooth-covering material. As far as the polycrystalline ceramic group is concerned, they have a fine-grained crystalline structure that provides strength and resistance to fracture but tends to have limited translucency. Furthermore, the absence of a glass phase makes polycrystalline ceramics difficult to abrade with hydrofluoric acid, requiring long times or higher temperatures [93].

Alumina has a high purity that can reach up to 99.5%, a high hardness (between 17–20 GPa), and a relatively high strength, since the elasticity value is 300 GPa which is much higher than all dental ceramics.

Zirconia exhibits more than twice the strength, a lower modulus of elasticity, and more brittleness compared to polycrystalline alumina. Pure zirconia is a very strong material that can accept pressures of more than 700 MPa, and for this reason it is used in permanent restorations. Its hardening process can be based on the stabilization of pure zirconium with specific agents/oxides such as yttrium, magnesium, and calcium [31,94].

To increase the biocompatibility and reduce the toxicity of orthopedic and dental implants, a thin layer of apatites is often used as a coating, as mentioned.

Bioactive glasses exhibit a hemostatic effect, and there is an increasing amount of research data showing that they also have an osteostimulating effect, since they can promote and accelerate bone formation at the cellular level. Bioactive glasses are successfully used to achieve bone regeneration in both maxillofacial and orthopedic surgery.

Resin-matrix ceramics include materials in which ceramic particles are at a huge advantage in terms of mass. They are materials containing mainly inorganic refractory compounds (>50% of their weight), including porcelain, glasses, ceramics, and glass–ceramics. Their development and production aim is (i) to obtain a material that more closely simulates the elastic modulus of dentin than traditional ceramics, (ii) to develop a material that is easier to derail and better suited than glass-matrix ceramics (e.g., synthetic ceramics of the lithium-disilicate family) or polycrystalline ceramics, and (iii) easy to repair with composite resin [95,96]. These materials can be divided into three subcategories, depending on their inorganic composition: (i) nanoceramic resin, highly aged and reinforced with about 80% in weight of nanoceramic material (combination of discrete nanoparticles of silicon and zirconia), (ii) glass–ceramic in a resin-interpenetrating matrix, and (iii) zirconia–silica ceramic in a resin-interpenetrating matrix adapted to different organic compounds and varying the weight percentage of the ceramic [97].

In addition to tailor-made mechanical properties, ceramics can easily achieve the desired shape and color, and for these reasons they are widely used in dentistry applications. Dental porcelain consists of a vitreous silicate matrix in which crystalline mineral salts are dispersed. The composition of the ceramic contains reduced quantities of metal oxides, which are used both as dyes to reproduce the color of natural teeth and to lower the melting temperature and increase the coefficient of thermal expansion [98,99,100,101]. Dental porcelain is used as a veneering material; to construct immobile frames such as metal–ceramic rims and bridges; to construct indirect aesthetic restorations, e.g., facades and inlays/overlays; and to create artificial teeth. During the last few years, the technology of all-ceramic systems has been developed to avoid the construction of fixed prosthetic devices (bridges and circles) made through a metal frame, e.g., entirely from ceramic biomaterials such as zirconia, alumina, and many others [102,103].

All of the objects created with the all-ceramic technique have significantly expanded the possibilities of their applications, especially in dentistry, making them more popular and allowing classic metal–ceramic restorations to be replaced progressively. This is a recurrent case because they combine high aesthetic performance and remarkable biocompatibility. High-strength glass–ceramic materials have the ability to improve aesthetic performance, as they bio-mimic the optical properties of hard dental tissues (enamel, dentin) in the best possible way. However, due to their low strength, they are used almost exclusively as cladding materials for high-strength ceramic frames. In essence, they do not differ significantly from conventional porcelain for metal–ceramic processing. However, the most appropriate choice of biomaterial for a customized implant must be sought by considering all the aesthetic and functional aspects for each patient. Several examples of bioceramic applications in dentistry are presented in Table 2.

### 3.4. Bioceramics for Bone-Tissue Engineering

Bone-tissue engineering is a multidisciplinary activity that implements mechanical-design principles in biomedical applications, primarily aiming at realizing volumetric and porous structures commonly known as biomimetic scaffolds. These elements are implanted in patients’ bodies to promote and guide bone-tissue regeneration in cases where large bone defects are present that cannot heal spontaneously. From a physiological point of view, it is important to point out the fact that cells can only randomly migrate to form two-dimensional layers, without any control of the shape to reconstruct to regenerate a damaged bone region [104]. Therefore, opportunely designed porous scaffolds act as biocompatible extracellular matrices, since they are engineered to support colonies of undifferentiated stem cells and to promote their differentiation and proliferation in a controlled way.

The general properties required to realize an optimal biomimetic scaffold for bone-tissue regeneration are (a) appropriate mechanical strength and stiffness to support the differentiating cells in load bearing during healing phases; (b) adequate surface properties to enable cell adhesion, differentiation, and osteointegration; (c) optimized topology and interconnectivity between pores to ensure cell migration, vascularization of the structure, and waste-material removal; (d) biocompatibility, intended as the capability of avoiding inflammatory or toxic responses in the implantation sites; (e) biodegradability, consisting of the process of being degraded and absorbed in a precise time period by the physiological environment of the implant; and (f) ease of fabrication into several shapes and dimensional scales [104].

As described in [105], the ceramic materials eligible for bone-tissue replacement can be divided into three main sets: structural ceramics, calcium phospates, and bioactive glasses. The first group includes alumina (Al_2_O_3_) and zirconia (ZrO_2_), which exhibit high hardness and high wear resistance, and this can be considered a problem if the stress-shielding effect is eventually induced in the implant [106]. This phenomenon occurs when a stiff scaffold material does not match the mechanical properties of the tissue to regenerate and, conversely, carries most of the imposed load, thus inhibiting, according to Wolff’s law, the natural growth and self-stiffening of bone tissue in the implantation site. Optimal scaffolds, in terms of mechanical properties, should match or be slightly higher than those of the hosting bone [107]. In the second group we can find hydroxyapatite (HA, [Ca_10_(PO_4_)_6_(OH)_2_]), which is the primary material constituting human bone, and tricalcium phosphate (TCP, [Ca_10_(PO_4_)_6_]). The latter is more biodegradable than the former when implanted in vivo and shows higher osteoinductivity [108] and osteoconductivity, which are, respectively, the capability of inducing osteogenesis during a bone-healing process such as a fracture and the tendency of a material to favor bone growth on its surface, which is a typical phase consequent to a bone Implant [109]. It is possible to enhance the bioactivity and biodegradability of synthetic HA by incorporating carbonates and divalent ions such as magnesium and strontium into the bulk material [110]. An innovative and sustainable source of raw biomaterials for highly biocompatible scaffolds, particularly indicated for the chemical synthesis of HA by using phosphorous compounds, is represented by the calcium carbonates that can be extracted from the by-products of the fish industry, in particular from mussel shells [111]. In the third group we can find many active compounds [112], such as calcium, sodium, and magnesium oxides, (CaO, Na_2_O and MgO, respectively) embedded in a silicon-dioxide (SiO_2_) bulk. These are highly biocompatible materials since they can dissolve in biological environments and can even enable chemical bonds with bone substrates. Nevertheless, a major drawback of glass bioceramics is the relatively low toughness of the glass bulk.

In order to reduce the negative effects caused by the brittleness of ceramic and to mimic the natural bone structure, which is essentially a biphasic material, composite scaffolds emerge as a suitable solution. They are obtained as a combination of a ceramic bulk and a natural or synthetic polymeric phase. The first component ensures high compressive strength and low degradation rates, whereas the second enhances the tensile strength and increases the overall toughness of the compound [110].

## 4. The Interaction Process between Bioceramic Materials and Bone Tissue

The three basic types of interfacial bonding are through physical, mechanical, and chemical forces. Mechanical bonds are a phenomenon that lies in the tension of resistance to inertia with the subsequent development of a specific interfacial structure. This interconnection is the basis for porous adjuncts and for traditional methods of bone repair through adjuncts such as screws, plates, and nails [113]. Bone ingrowth within the pores of a biomaterial can provide a specific and permanent support. However, mechanical strengths, which are determined by the mechanical bond with bone, are a problem area. The percentage of pore volume occupied by internal bone growth is almost never 100%, and therefore strength is reduced by the unoccupied volume fraction. Ossification occurs within the pores only if the pores are of the order of about 150 μm, thus only if the pore-size fraction is less than a critical size. Consequently, the interfacial strength is further reduced by the percentage of non-ossified pores. An additional parameter that reduces the mechanical–interfacial strength between porous biomaterials and bone is the reduction of the strength of the porous material, which is proportional to the pore size and the volume fraction of pores. According to the above, it is concluded that the mechanical–interfacial bond within a porous structure will result in 0.1–0.3 times the bond strength of a solid material, since ossification begins on and between the pores [114]. If there is a partial or complete inhibition of bone formation in areas of the porous interface with the simultaneous formation of a fibrous layer, then its strength at the interface tends to be reduced to 0.1–0.5 of the strength of normal bone. Conventional methods of skeletal-tissue recovery, which include screws, wires, etc., have disadvantages due to the low cross-sectional areas that carry a strong charge and the existence of fibrous membranes that reduce the stability of the interfacial bond [58]. In some cases, restoration is carried out through more than one plate, and their connection to the bone depends on the position of the plate and the quality of the connection. To achieve optimal strength and stiffness, orthopedic plates must be placed in such a way that the normal loading pattern can carry tensile load on each plate. In addition, the stiffness of the osteosynthesis is a transient phenomenon like the absorption of the bone around the threads. In particular, the anisotropy of the osseointegration is reduced by the intervention of the two plates restoring high levels of strength and stiffness of the fixation; however, the film formation and absorption problem is not completely eliminated [115].

Therefore, the mechanical bond presents significant limitations with consequent non-permanent restoration of the skeletal tissue. As for the physical bonds, which are activated through physical stimuli, they are not indicated as an effective bone-restoration technique, and chemical links are the most useful way to connect, through in situ induction. The types of chemical–interfacial bonds between biomaterials and tissues depend on the material and therefore on the molecules present on the surface to connect. For example, a possible bond that can occur is the direct covalent–ionic bond of the PO_4_^3-^ group simultaneously bonded to an organic component of the bone and a cation or oxygen ion on a ceramic surface [113]. The electrostatic chemical bond between a positively charged amino group (such as organic lysine, arginine, and hydroxylysine) and a negatively charged oxygen group of the ceramic surface can often occur. Regarding the weaker secondary chemical bonds, hydrogen bonds between a hydroxylated surface and a carboxyl group of an amino acid is a typical interaction [58]. The van der Waals forces emerging from a negatively charged surface can connect various groups of organic components. A more complex way of cross-linking is a composite ionic bond/van der Waals bond/hydrogen bond between a surface and the electrostatic charges in secondary collagen-binding sites. Another possible mode of cross-linking is the ionic–covalent bonding of an interfacial epitaxial crystal growth between a biomaterial surface and hydroxyapatite crystals, which are contained in the bone mass. Relatively, an epitaxial interface provides an ionic–covalent bond strength equivalent to that of the average of the two materials comprising the interface [114].

The interface mechanisms between bioceramics and tissues depend on the type of material to be used for bone restoration: (i) for a toxic implant the surrounding tissue dies; (ii) the non-toxic, bioinert material leads to the formation of a fibrous membrane on the tissue; (iii) the non-toxic, bioactive material develops interfacial bonds with the tissue; and (iv) the non-toxic, soluble material is replaced by the tissue [116,117]. Thus, the types of interfacial connection are recorded depending on the type of bioceramic. In the case of inert, microporous bioceramics, such as hydroxyapatite and alumina coatings, tissue growth occurs through the surface pores or through the implant. If the mobility in the environment of a porous implant is not satisfactory, the tissue tends to be damaged, the blood supply stops leading to tissue necrosis or inflammation, and a breakdown of interfacial stability between tissue and implant occurs [114]. Therefore, the open-porosity approach with defined pore size solves the problem in the case of fillers/prosthetic implants, but also in bioceramic metal coatings (hydroxyapatite layer on porous titanium metal surface). Regarding the category of resorbable bioceramics, the materials are gradually resorbed within a certain period and replaced by the natural tissue. The thickness of the interfacial interconnection of said bioceramics with human tissue is very thin or negligible. Absorbable bioceramics are an ideal biocompatible method in matters of tissue engineering; however, they do present some issues. Complications in the development of resorbable bioceramics arise from the fact that (i) the maintenance of strength and the stability of the interface during the period of degradation and replacement is not ensured, (ii) the rates of resorption differ significantly from the rates required for the repair of body tissues, and (iii) some biomaterials dissolve too quickly or too slowly. Furthermore, because a large amount of biomaterial can be replaced, it is essential that the bioresorbable implant consist only of metabolically acceptable and non-toxic substances. This criterion places significant limitations on the composition of resorbable biomaterials [58]. The degree of effectiveness of an implant depends on the thickness of the interfacial zone/layer between the material and the tissue. In the case where a biomaterial is nearly inert and the interface is not chemically or biologically bonded, relative movement and progressive growth of a non-adherent fibrous membrane over soft or hard tissue occurs. The movement at the biomaterial–tissue interface ultimately leads to a reduction of implant function, tissue deterioration, or destruction of the interface between them. The thickness of the developing fibrous membrane, created by the reaction of the body to the implant, varies depending on the material and the amount of relative movement [114]. The main types of bioceramic materials to be adopted for a specific interface are synthesized in Table 3.

## 5. Design and Fabrication of Ceramic Scaffolds

Porous bioceramics are aggregates of high mechanical strength commonly employed for scaffolding or structural bone-engineering bridges, as they help bone grow through their porous network. Porous bioceramics are widely used in tissue engineering to fabricate biomimetic scaffolds for bony tissue, often mimicking the morphology of natural lattices during the design phases. For example, it is possible to develop coral-like micro-structure materials that have a controlled pore size and adequate interconnectivity [1,118,119]. The hard-coral family of Porites (characterized by an average pore size of about 140–160 μm) and Goniopora (average pore size of about 200–1000 μm) are a model to imitate for the synthesis and production of porous materials such as α-alumina and calcium phosphates [120,121].

An ideal scaffold should possess a properly designed and optimized microgeometry that allows the formation of the largest amounts of bone in the shortest possible time. The design process of a personalized biomimetic scaffold starts from the acquisition of the real topologies of a severely damaged bone region that should be regenerated by means of a customized implant. This phase is usually carried out through computer tomography (CT) or magnetic resonance imaging (MRI) (Figure 2a); the biomedical data are then processed to isolate the bone regions from other organs and construct the outer surfaces of the anatomical district where the implant should be inserted (Figure 2b). A high-precision CAD model of the scaffold can then be created based on the actual boundary surfaces of the fracture area and the void region to fill with a porous structure in order to properly induce and guide the regeneration process of the bone tissue by following a shape as close as possible to that present before the damage (Figure 2c).

Biomimetic scaffolds can be modeled and engineered in their volumetric domains as a network of interconnected beams, thus forming structural lattices. From a geometrical point of view, such lattices can be classified in regular or periodic networks, commonly called honeycombs, and irregular or stochastic networks, often called foams. In the first case, the structure is obtained by the repetition of a unitary cell containing a precise distribution of nodes and beams. Several different shapes have been investigated through numerical simulation, such as the truncated cuboctahedron, the rhombic dodecahedron, the rhombicuboctahedron, and many others [122,123,124,125,126] (Figure 3). In the second case, the elements are distributed in space with a controlled degree of randomness, eventually following a predominant alignment to a specific loading direction [127]. The most influential geometrical-design variables, capable of determining correct mechanobiological stimuli, and hence precise physiological responses of undifferentiated cells adhering and proliferating on implanted scaffolds, are cell morphology and average pore size. It is worth noting that only narrow intervals for design parameters can trigger natural and sustainable osteogenesis and vascularization of the scaffolds [128]. In more detail, as reported in the literature, it is known that a porosity of 80% represents an optimal value to achieve good pore connectivity and at the same time sufficient mechanical resistance of the scaffold structure. Regarding the pore size, many guidelines can be found in scientific papers, even if the precise values are strictly dependent on the actual loading and environmental conditions of a defined anatomic district and should often be tailored to the single specific biomedical case. Recently, an ad hoc branch of the precision-medicine discipline has been developed, dedicated to the design and optimization of scaffolds for bone-tissue engineering. The minimum pore size ensuring cell colonization and osteoconduction can be assumed as 100 µm, whereas the mean pore size should be in the order of 300 µm to promote osteogenesis, vascularization, and oxygenation [110].

High-performance volumetric ceramic structures and, in particular, those based on irregular networks of beams can be algorithmically generated by topological optimization algorithms coupled with mechanical simulations. The optimal shape of a ceramic cluster of adjacent geometric unit cells is automatically computed through the maximization of a specific fitness function, which is typically represented by the volumetric fraction of mature bone tissue that is predicted to form inside the scaffold pores, knowing the loads and the constraints imposed on the considered construct [129,130]. Many mechano-regulation algorithms describing a bone-healing process were proposed and discussed for different cases in the literature in the last few decades, and thanks to the constant research in this field, the knowledge base is currently evolving and improving. Among others, two models have received great attention for bone-tissue-engineering applications: the model of Carter et al. and the model of Prendergast and Huiskes. In 1988, Carter et al. [131] hypothesized that the major factors inducing osteogenic or chondrogenic stimuli during the early stages of fracture healing are the cyclic stresses (distinguishing between tensile/compressive hydrostatic stresses and shear stresses) acting on the fracture site and the vascularity, which in turn affects the blood supply. In 1997, Prendergast and Huiskes [132] introduced a commonly accepted hypothesis, assuming that the biophysical stimulus responsible for osteogenesis is primarily composed of the magnitude of the octahedral shear strain acting on the considered substrate coupled with the rate of interstitial fluid flow. Therefore, from a physical point of view, the fracture domain can be modeled as a biphasic poroelastic material, and the results of the differentiation stage of the growing stem cells can be considered to be regulated by a combination of the presented variables. It is interesting to note that these kinds of generative approaches often return very complex topologies that can present size gradients for porosity distributions [131]. These shapes can be difficult to control in subsequent fabrication stages when using conventional techniques, in which is not always possible to precisely manage geometrical parameters such as pore sizing and morphology, void distribution, connectivity, and anisotropy [104].

A classic manufacturing process for realizing ceramic scaffolds is represented by the foaming method [110]. It consists of forming a porous ceramic scaffold starting from an aqueous suspension of pulverized raw materials (e.g., alumina, HA, or TCP) called slurries, which are then processed through template-assisted or template-free techniques (in the latter, a foaming agent creates gaseous products when heated, as in the case of porous-alumina processing [133]) and then consolidated by means of thermal treatments, as depicted in Figure 4a.

An evolution of foaming techniques is represented by freeze-casting, which is based on the controlled freezing of ceramic slurries, and it offers the chance to control the pore shape and anisotropy of scaffolds (the latter is an interesting feature for creating load-bearing structures) by managing the parameters governing the advancing cooling front and the properties of additives and solvents mixed into the slurries. Furthermore, porous ceramic surfaces can be realized by mixing soluble metal or salt particles on a surface. The pore size and the porosity of the microstructure are always in agreement with the size and shape of the impurity particles, which are removed by using a suitable corrosive agent. The porous layer produced with this technique is an integral part of the denser ceramic phase constituting the bulk material. Therefore, to obtain an adequate porous network given a certain porous-ceramic composition, it is important to know that the initial conditions and the materials used in the forming process are strictly related to the characteristics obtained in the final products, making it necessary to accurately control the choices regarding the correct materials and the key parameters of production phases.

Nowadays, the most efficient approach for manufacturing highly optimized and complex topologies in a repeatable way is represented by additive-manufacturing (AM) techniques, also known as laser-based solid freeform fabrication (SFF). Different from classical ceramic-scaffold-fabrication techniques, in which there is little or no control of resulting topologies or an additional template is required, solid freeform technologies offer a wider range of design opportunities, from macro scales to microscopic features. Additive manufacturing is currently considered a well-consolidated practice for biomedical application, since it perfectly integrates with digital imaging and processing, it enables tailoring and customization of biomedical solutions, and it requires no tooling and minimal setup time [134]. The starting point in additive approaches is a CAD model of the component to create, which is typically processed by a slicing software that computes the intersection of the solid regions with a series of parallel planes and elaborates the geometrical paths necessary for the machines to create the object through a layer-by-layer strategy. Among the additive techniques for fabricating scaffolds, selective laser sintering (SLS) of ceramic powders with bioactive nanofillers plays a role of great relevance [135,136]. Typically, in the SLS process, a thin layer of ceramic powder is repeatedly spread onto a flat surface by a moving a roller and then scanned with a CO_2_ laser beam following the previously elaborated path to additively sinter layers of powders constituting stacked cross-sections of the scaffold to create. The machine setup of the SLS process is illustrated in Figure 4b.

## 6. Conclusions

In this work, the characteristic features of ceramic materials and their possible applications in biomedical fields are presented. The physical and mechanical properties of bioceramic materials are analyzed, with particular attention to their high strength, wear resistance, and brittleness, with the latter being the main drawback when bioceramic implants are subjected to external loads. Regarding the chemical properties, the biocompatibility with human tissues and the capability of promoting osteointegration are outlined. A wide range of commonly used and advanced bioceramic materials is described, and several biomedical procedures related to the application of bioceramic materials in dentistry and bone-tissue engineering are discussed, thus giving further details on the interactions between implant interfaces and bone tissue and on biomimetic-scaffolds design.

## Figures and Tables

**Figure 1 jfb-14-00146-f001:**
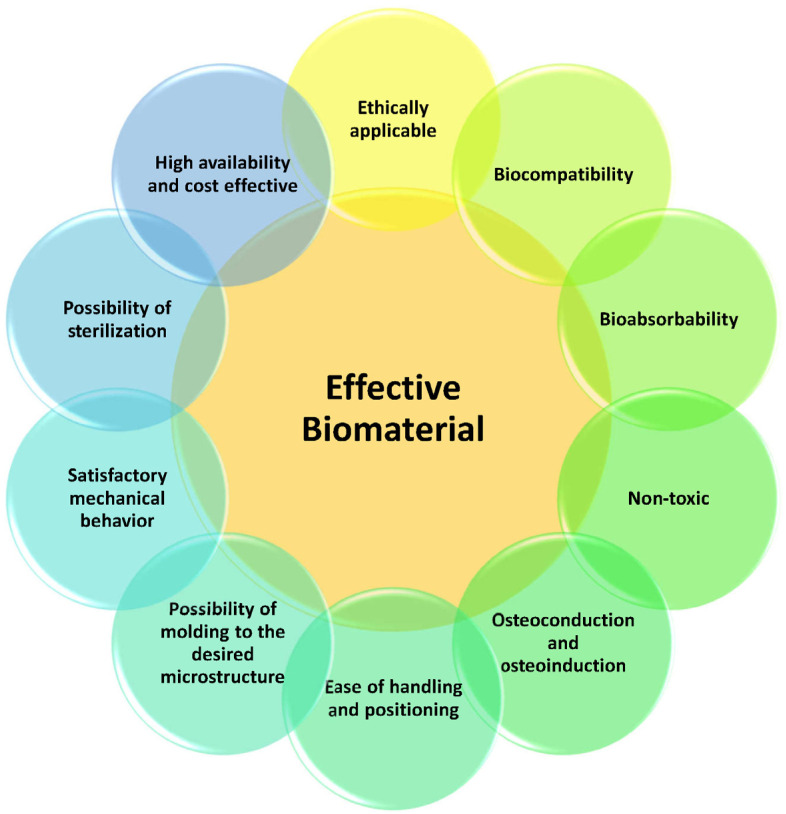
The reasoning for an effective biomaterial to be employed in the medical field.

**Figure 2 jfb-14-00146-f002:**
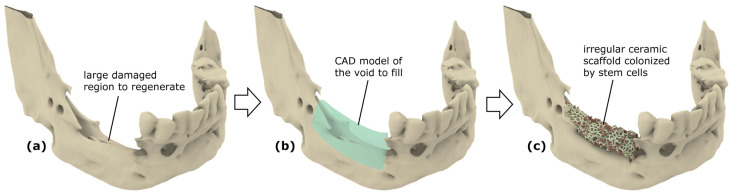
(**a**) Three-dimensional model of the damaged bone acquired through biomedical imaging; (**b**) CAD model representing the boundary surfaces of the volume to fill (in green); (**c**) customized ceramic scaffold implanted in the region to regenerate.

**Figure 3 jfb-14-00146-f003:**
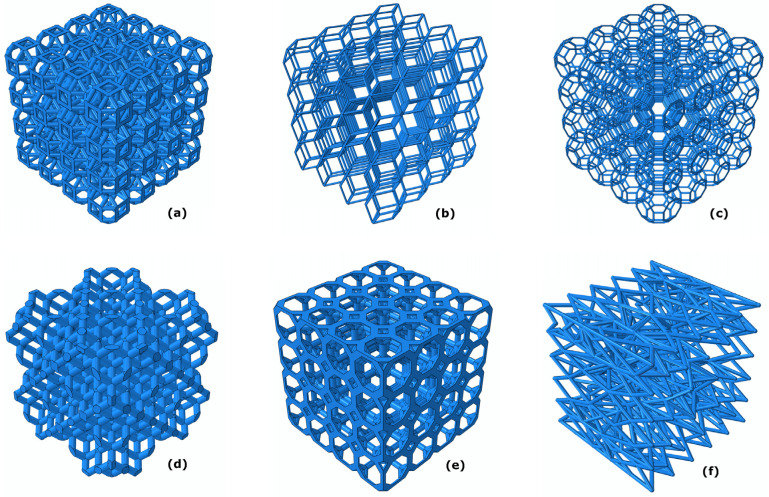
Regular (**a**–**e**) and load-adapted irregular (**f**) scaffolds have been designed and optimized based on mechanobiological criteria and, in particular, by implementing the mechano-regulation algorithm developed by Prendergast and Huiskes. Regular scaffolds include different unit-cell geometries: (**a**) rhombicuboctahedron; (**b**) rhombic dodecahedron; (**c**) truncated cuboctahedron; (**d**) diamond; € truncated cube.

**Figure 4 jfb-14-00146-f004:**
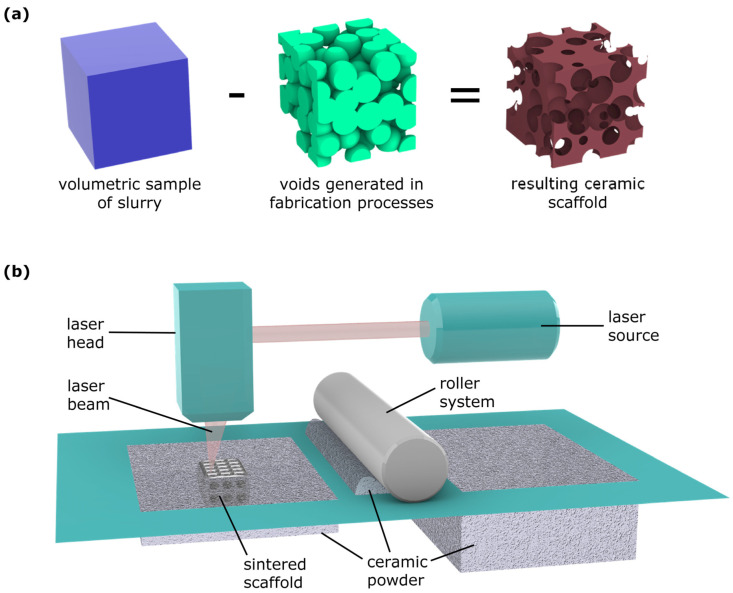
Two principal fabrication techniques for the realization of ceramic scaffolds: (**a**) three-dimensional model of a ceramic-scaffold sample obtained through templates; (**b**) scheme of selective laser sintering for a ceramic scaffold.

**Table 1 jfb-14-00146-t001:** Properties and biomedical applications of the main bioceramic materials [47,48,49,50,51,52,53,54,55,56].

Material	Young’s Modulus (GPa)	Compressive Strength (MPa)	Density (g/cm^3^)	Bioactivity	Applications
Alumina	380	4000	>3.9	Inert	Orthopedics, load-bearing applications, dentistry
Zirconia	150–200	2000	6.0	Inert	Orthopedics, load-bearing applications, dentistry
Porous hydroxyapatite	70–120	600	3.1	Bioresorbable	Dentistry, coatings, scaffolds
Tricalcium phosphate	120–160	540	3.1	Bioresorbable	Dentistry, scaffolds
Bioactive glasses	75	1000	2.5	Bioactive	Dentistry, spinal surgery

**Table 2 jfb-14-00146-t002:** Examples of bioceramic materials for dental and periodontal surgery.

Type of Intervention	Bioceramic Material
Surface coatings (dental and maxillofacial implants)	HA, bioactive glass, bioactive glass–ceramic
Dental implants	Alumina, HA, bioactive glass
Periodontal surgery	HA, HA-PLA, calcium and phosphorous salts, bioactive glass
Implants with alveolar-ridge augmentation	Alumina, HA, HA-PLA, HA–autogenous bone composite, bioactive glass
Coatings for tissue growth	Alumina, HA

**Table 3 jfb-14-00146-t003:** Types of connection with bone tissue and suitable bioceramic materials.

Interface Type	Bioceramic Material
Morphological interconnection (bone growth through superficial imperfections)	Dense, non-porous, partially inert ceramics (alumina)
Biological interface (bone growth through mechanical linkages with the implant)	Porous, inert ceramics (alumina, HA coatings)
Bioactive interface (direct bone bonding through chemical ties)	Dense, non-porous, active ceramics (HA, bioactive glass, bioactive glass–ceramic)
Gradual bone replacement	Dense, porous or non-porous, resorbable ceramics (HA, tricalcium phosphate)

## Data Availability

Data sharing not applicable.

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
