# Peer review of "Ceramic Materials for Biomedical Applications: An Overview on Properties and Fabrication Processes"

_jfb, 2023, doi:10.3390/jfb14030146_

Round 1

Reviewer 1 Report

"Review on Ceramic Materials: An Overview" is not well organized, and my comments are below.

1.      Figure 1 is not discussed in the text. Why do the authors use the following terms: design, manufacturing, and standardization.

2.      Line 49, Provide references for your claim.

3.      Figure 2 – I am not understanding what the authors want to say. Is this the entire process that biomaterials must go through? why the biomaterial needs to be porous.

4.      The title of the manuscript is about ceramic materials. But nothing was discussed about ceramics in the introduction part. The review should be more focused than just explaining basic terms.

5.      Physical properties of ceramic materials too much basic information. not required. This information is available on Wikipedia. Please revise.

6.      Line 104: The authors quoted different applications. I request the authors revise the article only for biomedical applications.

7.      Please discuss more on: why ceramics are needed in biomedical industries; what is the key property one must look into while selecting ceramics; how ceramics interact with the human body; what are the adverse effects and how to modify ceramic materials, the use of ceramic composites, etc.

8.      Chemical properties: If the author is discussing different ceramic materials under one title, first state the need or problem and then discuss the issue. There is no flow, and it is difficult to understand what the authors want to say about this subject.

9.      Mechanical properties of ceramics—more general information about ceramics—What is the need for mechanical properties in biomedical applications? particularly in joints, wear counterparts, and dental applications.

10.   Ceramic materials for dentistry applications – highlight the materials used in dentistry applications and their physical and chemical properties in table form.

11.   Figure 3: The majority of the ceramics listed are not discussed in the text (for example, TCP, HAp, and bioactive glasses).

12.   Design and fabrication of ceramic scaffolds – following parts are good and focused. It would be better if the author change the title to "ceramic scaffolds and its fabrication processes for biomedical applications" and alter the introduction. Because the title is not reflecting the actual review.

Author Response

Bari (Italy), February 14th 2023

Ob: Submission of the revised article jfb-2191729 “Ceramic Materials: an Overview on Properties and Biomedical Applications” to Journal of Functional Biomaterials

Dear Editor,

we received your communication where you informed us that the Reviewers recommend reconsideration of the manuscript jfb-2191729  - “Ceramic Materials: an Overview on Properties and Biomedical Applications”, Authors: L. Vaiani, A. Boccaccio, A.E. Uva, G. Palumbo, A. Piccininni, P. Guglielmi, S. Cantore, L. Santacroce, I.A. Charitos, and A. Ballini.

According to the comments provided by Reviewers, we made substantial changes in the article and added several new sections. In detail:

  • we improved the text flow and the paragraphs subdivisions, to obtain a better consequentiality;
  • we re-edited the figures, where needed;
  • we inserted several tables synthesizing the concepts discussed in the text;
  • we added a section about nanocomposites and ceramic coatings, and a section about the interactions between bioceramic materials and living bone tissues;
  • we checked for minor issues and removed all the off-topic sentences regarding non-ceramic and non-biomedical arguments.

Below we have reproduced the Reviewers’ comments in bold typeface and given our reply in italics. Where necessary, we have included parts of the revised manuscript in our replies in inverted commas.

Please let us know if the revised paper now satisfies all requirements for publication.

Should you have any questions, please do not hesitate to contact me.

Thank you very much in advance for your attention and courtesy.

Yours Sincerely,

Antonio Boccaccio, Stefania Cantore and Andrea Ballini

REPLY TO REVIEWER 1

"Review on Ceramic Materials: An Overview" is not well organized, and my comments are below.

  1. Figure 1 is not discussed in the text. Why do the authors use the following terms: design, manufacturing, and standardization.
  2. Line 49, Provide references for your claim.
  3. Figure 2 – I am not understanding what the authors want to say. Is this the entire process that biomaterials must go through? why the biomaterial needs to be porous.

We totally agree with the Reviewer about these points. Since Figure 1 and Figure 2 were unclear, they have been removed and re-edited, respectively. The same revision was conducted on all the unclear sentences present in the text. Concerning point 2, we removed from the text all the parts describing non-ceramic materials.

  1. The title of the manuscript is about ceramic materials. But nothing was discussed about ceramics in the introduction part. The review should be more focused than just explaining basic terms.
  2. Please discuss more on: why ceramics are needed in biomedical industries; what is the key property one must look into while selecting ceramics; how ceramics interact with the human body; what are the adverse effects and how to modify ceramic materials, the use of ceramic composites, etc.

We sincerely thank the Reviewer for these comments that helped us to improve the quality of the manuscript. We defined a paragraph in the introduction part, after Figure 1, introducing the essential features of bioceramics making them a suitable solution for biomedical issues. In the following part (“Physical and chemical properties of ceramics”), and in other parts throughout the manuscript (e.g. “Advanced Bioceramics”) we focused about bioceramic nanocomposites, bioactive coatings and related biomedical applications. We report here, as an example, one part we added in the manuscript for addressing the highlighted points:

From “1. Introduction” (please, see from line 60 to line 73 of the revised manuscript):

“[…]. Compared to other biomaterials such as metals or polymers, bioceramics possess a unique combination of properties, such as i) high intrinsic strength: materials as alumina or zirconia show great mechanical properties as high wear resistance and low coefficient of friction, making them suitable for usage in high-stress applications as artificial joints or dental implants; ii) biocompatibility: bioceramics are, in general, compatible with human tissues, reducing the risk of adverse reactions or inflammations and some bioceramics in particular, such as hydroxyapatite or bioactive glasses, show bioactive behaviors that can promote tissue regeneration and osteointegration; iii) versatility: bioceramics can be modelled to precise shapes and their compositions can be tailored to enhance specific properties. All of these features make bioceramics an adequate solution for a wide variety of biomedical issues. Research on ceramic biomaterials is developing rapidly, finding new key applications in medicine and biotechnology, especially towards their usage as load-bearing parts, joint replacements, fillers, veneering materials, drug delivery platforms, and biomimetic scaffolds.”

  1. Physical properties of ceramic materials too much basic information. not required. This information is available on Wikipedia. Please revise.
  2. Mechanical properties of ceramics—more general information about ceramics—What is the need for mechanical properties in biomedical applications? particularly in joints, wear counterparts, and dental applications.

We reduced the section about the general properties of ceramics and distributed more application-specific discussions about physical properties throughout the manuscript. Some examples are reported:

From “Bioceramics for general applications” (please, see from line 168 to line 173 of the revised manuscript):

“[…]. Alumina (Al2O3) and zirconia (ZrO2) are the two most important ceramic oxides for biomedical purposes, which are used for damaged bone tissue or joint repair and replacement, as in case of total hip and knee arthroplasty, due to their excellent wear resistance and biocompatibility. They are inert materials, but can be used in combination with other materials, such as biodegradable polymers, to deliver drugs and promote tissue regeneration.”

From “Advanced bioceramics” (please, see from line 277 to line 282 of the revised manuscript):

“[…]. The characteristic trait of nanocomposite materials is their ability to combine properties and functionalities that are out of reach for traditional materials. By incorporating nanoparticles into a ceramic matrix (e.g. by adding organic molecules, carbon nanotubes, graphene, nanoscale ceramics, proteins or even DNA to bioceramics or bioglasses), it is possible to create materials with improved mechanical strength, biocompatibility and osteoconductivity. […]

(please, see from line 334 to line 338 of the revised manuscript):

[…]. Another outstanding development in bioceramic field has been the creation of bioactive coatings. The surfaces of metal or ceramic implants can be coated with ceramic layers, bioactive molecules or antimicrobial agents for preventing the risk of infections and promoting tissue regeneration, wound healing, and osteointegration with the surrounding tissues, thus obtaining an effective functional material.

(please, see from line 341 to line 344 of the revised manuscript):

[…] The use of hydroxyapatite as a coating on orthopedic and dental metallic implants combines the advantages of metallic materials in terms of mechanical properties with the excellent biocompatibility and bioactivity of hydroxyapatite. In fact, this material coupling is very popular.”

  1. Line 104: The authors quoted different applications. I request the authors revise the article only for biomedical applications.

We accurately removed from the text all the parts describing non-biomedical applications.

  1. Chemical properties: If the author is discussing different ceramic materials under one title, first state the need or problem and then discuss the issue. There is no flow, and it is difficult to understand what the authors want to say about this subject.

We completely re-arranged the subdivisions in different paragraphs and revised the whole text flow, in order to obtain logical discussions about the presented topics. Regarding the chemical properties reported in section 2 (“Physical and chemical properties of ceramics”), we focused on bioactivity of ceramic materials.

From “Physical and chemical properties of ceramics” (please, see from line 98 to line 112 of the revised manuscript):

“[…]. Depending on the chemical properties (molecular bioactivity when interacting with human organisms), ceramic biomaterials can be classified as: (a) inert, (b) low or medium surface activity, (d) bioresorbable (adsorbable) ceramics. The choice of the material to be used (inert, bioactive or bioresorbable) depends on the function to be accomplished in every specific application. Inert bioceramics as alumina (Al2O3), do not promote the connection with living tissues, can withstand low-pH environments for thousands of hours, and possess high chemical inertia, which in turn means that they require a long time until stable connections between implants and tissues are established. Once implanted, they are surrounded by a network of fibrous connective tissue of varying thickness, which holds the implant and at the same time isolates it from adjacent tissues. Therefore, due to their high biocompatibility and mechanical strength, they are designed for permanent implants [23]. The low and medium activity materials, in addition to binding to specific proteins, can also release ions, thus promoting the integration of implants to living tissues [24]. Finally, the bioabsorbable ceramics are destined to remain until the regeneration of the new tissue where they are inserted, occurs.”

  1. Ceramic materials for dentistry applications – highlight the materials used in dentistry applications and their physical and chemical properties in table form.

We sincerely thank the Reviewer for helping us to improve the readability of the work. We described the primary bioceramics used for dentistry and other applications and summarized the related properties in three different tables included in the manuscript, in “Bioceramics for general applications” (please, see from line 152 of the revised manuscript), “Bioceramics for dentistry applications” (please, see from line 369 of the revised manuscript), and “4. The interaction process between bioceramic materials and bone tissue” (please, see from line 512 of the revised manuscript).

  1. Figure 3: The majority of the ceramics listed are not discussed in the text (for example, TCP, HAp, and bioactive glasses).

We agree with the Reviewer about Figure 3, so we removed it and revised the text in order to gain a wider discussion about the main bioceramics employed in biomedical field. We expanded the descriptions and added more references in sections “Bioceramics for general applications” (please, see from line 152 of the revised manuscript) and “Bioceramics for dentistry applications” (please, see from line 369 of the revised manuscript).

  1. Design and fabrication of ceramic scaffolds – following parts are good and focused. It would be better if the author change the title to "ceramic scaffolds and its fabrication processes for biomedical applications" and alter the introduction. Because the title is not reflecting the actual review.

We sincerely thank the Reviewer for having pointed out this weakness in the work. We changed the title to: “Ceramic Materials for Biomedical Applications: an Overview on Properties and Fabrication Processes”.

Reviewer 2 Report

I agreed to review the article because of the title. It sounds nice and attractive and in the area of my scientific interest. I made some comments in the pdf file. I think that the authors have low level of understanding what ceramics is. Sometimes authors forgot about the title of the article.  Probably you should discuss the text of the article with someone who knows the definition of ceramics. Probably the list of co-authors then can be changed. You wrote sometimes about properties of ceramics at all, without any connection with the possible usage as biomaterials (strength at 1400oC for example). The title of article is very attractive. I think that you can work on the text and resubmit the article. Probably the title also can include information about bioceramics production. Probably you can exclude description of bioceramics production and do not change the title.

Author Response

Bari (Italy), February 14th 2023

Ob: Submission of the revised article jfb-2191729 “Ceramic Materials: an Overview on Properties and Biomedical Applications” to Journal of Functional Biomaterials

Dear Editor,

we received your communication where you informed us that the Reviewers recommend reconsideration of the manuscript jfb-2191729  - “Ceramic Materials: an Overview on Properties and Biomedical Applications”, Authors: L. Vaiani, A. Boccaccio, A.E. Uva, G. Palumbo, A. Piccininni, P. Guglielmi, S. Cantore, L. Santacroce, I.A. Charitos, and A. Ballini.

According to the comments provided by Reviewers, we made substantial changes in the article and added several new sections. In detail:

  • we improved the text flow and the paragraphs subdivisions, to obtain a better consequentiality;
  • we re-edited the figures, where needed;
  • we inserted several tables synthesizing the concepts discussed in the text;
  • we added a section about nanocomposites and ceramic coatings, and a section about the interactions between bioceramic materials and living bone tissues;
  • we checked for minor issues and removed all the off-topic sentences regarding non-ceramic and non-biomedical arguments.

Below we have reproduced the Reviewers’ comments in bold typeface and given our reply in italics. Where necessary, we have included parts of the revised manuscript in our replies in inverted commas.

Please let us know if the revised paper now satisfies all requirements for publication.

Should you have any questions, please do not hesitate to contact me.

Thank you very much in advance for your attention and courtesy.

Yours Sincerely,

Antonio Boccaccio, Stefania Cantore and Andrea Ballini

REPLY TO REVIEWER 2

I agreed to review the article because of the title. It sounds nice and attractive and in the area of my scientific interest. I made some comments in the pdf file.

I think that the authors have low level of understanding what ceramics is. Sometimes authors forgot about the title of the article. Probably you should discuss the text of the article with someone who knows the definition of ceramics. Probably the list of co-authors then can be changed. You wrote sometimes about properties of ceramics at all, without any connection with the possible usage as biomaterials (strength at 1400oC for example).

We sincerely thank the Reviewer for the very useful comments that helped us to improve the quality of the manuscript and that were addressed in the revised version.

Concerning the definition of ceramic material, we adopted as a main reference the book from William D. Callister, Jr. “Material Science and Engineering: an Introduction”, which is a commonly known reference in the field. The book has been included in the list of references in the revised manuscript.

We completely re-arranged the subdivisions in different paragraphs and revised the whole text flow, in order to obtain logical discussions about the presented topics. We accurately revised all the statements about ceramic materials and added new references. We removed from the text all the parts describing non-ceramic materials and non-biomedical applications. We reduced the section about the general properties of ceramics and distributed more specific discussions about biomedical applications throughout the manuscript. Furthermore, we inserted three tables synthesizing some concepts discussed in the text and we added new paragraphs and new complete sections, as: “Bioceramics for general applications” (please, see from line 152 of the revised manuscript), “Bioceramics for dentistry applications” (please, see from line 369 of the revised manuscript), and “4. The interaction process between bioceramic materials and bone tissue” (please, see from line 512 of the revised manuscript). Some parts added in the manuscript and addressing the highlighted points are reported in the following.

From “1. Introduction” (please, see from line 60 to line 73 of the revised manuscript):

“[…]. Compared to other biomaterials such as metals or polymers, bioceramics possess a unique combination of properties, such as i) high intrinsic strength: materials as alumina or zirconia show great mechanical properties as high wear resistance and low coefficient of friction, making them suitable for usage in high-stress applications as artificial joints or dental implants; ii) biocompatibility: bioceramics are, in general, compatible with human tissues, reducing the risk of adverse reactions or inflammations and some bioceramics in particular, such as hydroxyapatite or bioactive glasses, show bioactive behaviors that can promote tissue regeneration and osteointegration; iii) versatility: bioceramics can be modelled to precise shapes and their compositions can be tailored to enhance specific properties. All of these features make bioceramics an adequate solution for a wide variety of biomedical issues. Research on ceramic biomaterials is developing rapidly, finding new key applications in medicine and biotechnology, especially towards their usage as load-bearing parts, joint replacements, fillers, veneering materials, drug delivery platforms, and biomimetic scaffolds.”

From “Advanced bioceramics” (please, see from line 334 to line 338 of the revised manuscript):

“[…]. Another outstanding development in bioceramic field has been the creation of bioactive coatings. The surfaces of metal or ceramic implants can be coated with ceramic layers, bioactive molecules or antimicrobial agents for preventing the risk of infections and promoting tissue regeneration, wound healing, and osteointegration with the surrounding tissues, thus obtaining an effective functional material.”

The title of article is very attractive. I think that you can work on the text and resubmit the article. Probably the title also can include information about bioceramics production. Probably you can exclude description of bioceramics production and do not change the title.

We sincerely thank the Reviewer for pointing this out. To address this issue, we changed the title to: “Ceramic Materials for Biomedical Applications: an Overview on Properties and Fabrication Processes”.

Other comments from PDF:

Line 34. Let's talk about patients, not about us.

The correction required by the Reviewer was implemented in the revised manuscript (please, see from line 33 to line 36 of the revised manuscript)

“[…]. The main purpose of biomaterials is to support the healing or the replacement of an organ in a human body that has been altered by a disease or an accidental event and to successfully restore functions and sometimes aesthetics features without endangering human life [2].”

Figure 1. Sterilization before standardization? Sterilization is a obvious part of procedure of testing in animals and patients. What is the reason to stress this point? I think that this scheme is a mix of two schemes. One exists for new created material, another for material which have permission to be used in patient treatment (long term monitoring in patients). Is this article about devises or about materials?

The reviewer is absolutely right. Figure 1 was unclear, so we removed it from the manuscript and revised the text, accordingly.

The title of the article is: Ceramic Materials: an Overview on Properties and Biomedical Applications. 53-59 - are not about ceramic materials.

Lines 53 to 59 were removed from the manuscript according to the suggestion of the Reviewer.

Figure 2. What does arrows mean in this scheme? Arrows have to have meanings. But they have not meanings here. All mentioned features deal with the bioceramics. Some of them connected with making choice. And some of them are about production and composition. Some of them about way of using. I think the boxes should be grouped according in another way then using arrows. If you give this scheme - all points should be discussed.

Figure 2 has been edited and the arrows were removed. The items included in the bubbles of the new diagram were briefly described in the revised manuscript. Please, see lines 47-57.

“The application of a specific biomaterial is driven by the necessary composition, material properties, structure, and the triggering of desired in vivo reactions, in order to perform a precise function. Furthermore, for the usage in medical field, researchers need to make attention to bioethics, biocompatibility, bioabsorbency and toxicity [7,8]. A possible categorization of different peculiar properties for a biomaterial to be employed in medical applications in order to maximize functional results is presented in Figure 1. As an example, in tissue engineering applications, biomaterials must be capable of being modelled to the shape and size proper of the section of the organic part to be replaced, and the surface of the replacement part must possess a precise roughness, for inducing cell adhesion and to favor biological integration with the tissues or the skeleton, while the inner topology of the replacement part must present a porous bulk, as described in the following sections.“

Lines 92 to 94. The topic was not disclosed. This part of the article discuss the different kind of ceramic materials without deep understanding of what ceramics as material is. I can not agree with this definition. Ceramics are inorganic materials (non-metallic - yes) prepared from powders using sintering (high-temperature sintering). Cements are chemically bonded materials.

Lines 99-100. Ceramics can not be find in nature. It is artificial material. Clays - is not an example of ceramics. Clays - are starting product for ceramic production.

Concerning the definition of ceramic material, we adopted as a main reference the book from William D. Callister, Jr. “Material Science and Engineering: an Introduction”, which is a commonly known reference in the field. The book has been included in the list of references in the revised manuscript.

Line 102. False statement

The statement was removed from the manuscript.

Line 117. this is about composite with metal matrix - it is not ceramics.

The entire paragraph was re-edited and the statement regarding the composite materials with metal matrix was removed from the manuscript.

Lines 123-126. This is not about ceramics. Ceramics - polycrystal or glass-crystal material. Especially polycrystal those which are used in hip or knee treatment. You mixed oxide as chemical substance or crystal and ceramics as material

The entire paragraph was re-edited and this statement was removed from the manuscript.

Line 131. Can you explain about surface of oxide? Material can have surface.

The Reviewer is absolutely right. The statement was unclear, so it was changed to “[…]. On the surfaces of the implants realized with these materials are commonly found free hydroxyl radicals (-OH) which interact with body fluids, providing a lubricating layer around implants. […]”. Please, see line 176 of the revised manuscript.

Line 134. It is false statement. Any ceramic material is polycrystalline according definition. Alumina can be prepared in form of monocrystal or in form of polycrystal. To be Al2O3 is not enough to be treated as polycrystalline.

The entire paragraph was re-edited and this statement was removed from the manuscript.

Line 140. What about alumina? No grinding for alumina? Line 142. Do you think that only phase stabilization play a role?

The entire paragraph was re-edited and this statement was removed from the manuscript.

Line 185. Nice title. The topic was not disclosed.

We completely re-arranged the subdivisions in different paragraphs and revised the whole text flow, in order to obtain logical discussions about the presented topics. Regarding the chemical properties reported in section 2 (“Physical and chemical properties of ceramics”), we focused on bioactivity of ceramic materials and we reported several discussions throughout the manuscript as in the parts: “Advanced Bioceramics” where we focused about bioceramic nanocomposites, bioactive coatings and related biomedical applications; and in Section “4. The interaction process between bioceramic materials and bone tissue” where we discussed about biochemical/biological interactions between the ceramic implant and the bone tissue.

Lines 202-203. Ceramics based on HA is not bioabsorbable.

We revised the entire paragraph and removed HA from the list of bioresorbable ceramics in the text, since HA typically shows long-term biodegradability. We only  indicated in table 1 “bioresorbable” for porous HA, since we cannot state that porous HA is strictly non-bioresorbable, according to the studies reported in:

Takahiro Goto, Tatsuyoshi Kojima, Takuo Iijima, Satoshi Yokokura, Hirotaka Kawano, Aiichiro Yamamoto, Koichi Matsuda, “Resorption of synthetic porous hydroxyapatite and replacement by newly formed bone”, Journal of Orthopaedic Science, Volume 6, Issue 5, 2001, Pages 444-447, ISSN 0949-2658, https://doi.org/10.1007/s007760170013.

In the last paragraph the Authors of the paper above state:

“In conclusion, HA is biodegradable, although it takes a long time to be biodegraded. The biodegradability of HA seems to be influenced by sintering temperature, porosity, and pore diameter.”

Another remarkable discussion we followed as a reference is reported in the section “4. Dissolution and Reprecipitation as Bone” of the work:

Eliaz N, Metoki N. Calcium Phosphate Bioceramics: A Review of Their History, Structure, Properties, Coating Technologies and Biomedical Applications. Materials (Basel). 2017 Mar 24;10(4):334. doi: 10.3390/ma10040334. PMID: 28772697; PMCID: PMC5506916.

Line 211-212. Listed hydroaluminosilicates are minerals of clay. After firing (the must stage of ceramic production) they lose the water and became alumina silicates. Porcelain including mullite (alumina silicate) are in the planet for ages, not resorbable.

The entire paragraph was re-edited and this statement was removed from the manuscript.

Lines 218-222. This information can help if you plan to synthesize brushite in form of powder or cement stone. Tetracalcium phosphate is high temperature phase, very toxic for living tissue.

The statements regarding tetracalcium phosphate were removed, according to the precious suggestion of the Reviewer, from the paper as it deals with the biomedical applications of ceramics. We sincerely thank Her/Him for pointing this aspect out.  

Lines 240-242. The title of the article is: Ceramic Materials: an Overview on Properties and Biomedical Applications. Not about materials in general.

The statement was better focused on the ceramic materials which are the actual objective of the review as well as references specifically regarding ceramics were cited in the revised manuscript. Please, see lines 269-272 of the revised manuscript.  

Line 257. What is it?

We thank the Reviewer for pointing out this unclear aspect which was clarified in the revised manuscript. Please, see lines 126-131.

“These weak responses occur because at micro scales exists a wide variety of imperfections of different size and geometry, such as internal pores, micro-cracks, grain misalignments, impurities, microscopic notches and so on [17]. These defects are usually formed in the production process and are generally caused by thermal gradients induced by heat cycles. In general, ceramics with finer microstructures show better mechanical properties.”

Lines 273-274. What kind of ceramics do not react with environment? Those that can be absorbed by living tissue during implantation in regeneration process?

The entire paragraph was re-edited and this statement was removed from the manuscript.

Lines 283-284. Do not we forget that we are here discuss the biomaterials? Transhumanism on its way? 1400 oC - for which living organism is important?

Line 288. Why important for biomaterials?

All the references to the applications of ceramic materials falling outside the biomedical field were removed from the manuscript. We sincerely thank the Reviewer for helping us to better focus the scope of the review paper.

Line 383. Correct material choice consists in what? Figure 3. Comments of arrangement of materials are needed.

We thank the Reviewer for pointing out this unclear aspect which was clarified in the revised manuscript. Please, see lines 451-454.

“[…]. However, the most appropriate choice of the biomaterial to be chosen for a customized implant must be sought, considering all the aesthetic and functional aspects for each patient. Several examples of bioceramic applications in dentistry are presented in Table 2.”

Figure 3 was unclear, so we removed it from the manuscript and revised the text, accordingly.

Line 442. This point is not reflected in title of article.

We sincerely thank the Reviewer for pointing out this weakness in the work. We changed the title to: “Ceramic Materials for Biomedical Applications: an Overview on Properties and Fabrication Processes”.

Line 452. These are not ceramics.

We removed all the references concerning non-ceramic materials.

Figure 6. First line of the figure is not about using foaming agent. This is about sacrificed template.

We thank the Reviewer for pointing out this slip. The figure was edited as well as its caption.

All the comments included in the PDF document were accurately taken into account and fully addressed in the revised manuscript. Again, we sincerely thank the Reviewer for the very appropriate suggestions.

Reviewer 3 Report

Development of ceramic nanocomposite is one of the latest trends for creating materials with outstanding properties. Hence, it is suggested to add a brief paragraph on ceramic nanocomposites developed for biomedical applications.

Explain how bioactive molecules are coated on the surface of the ceramic implants to develop functional bioceramics. Also, to reduce the risk of implant related infections or to prevent failed implants, surface of the implants are coated with antimicrobial agents. Cite few important studies.

Author Response

Bari (Italy), February 14th 2023

Ob: Submission of the revised article jfb-2191729 “Ceramic Materials: an Overview on Properties and Biomedical Applications” to Journal of Functional Biomaterials

Dear Editor,

we received your communication where you informed us that the Reviewers recommend reconsideration of the manuscript jfb-2191729  - “Ceramic Materials: an Overview on Properties and Biomedical Applications”, Authors: L. Vaiani, A. Boccaccio, A.E. Uva, G. Palumbo, A. Piccininni, P. Guglielmi, S. Cantore, L. Santacroce, I.A. Charitos, and A. Ballini.

According to the comments provided by Reviewers, we made substantial changes in the article and added several new sections. In detail:

  • we improved the text flow and the paragraphs subdivisions, to obtain a better consequentiality;
  • we re-edited the figures, where needed;
  • we inserted several tables synthesizing the concepts discussed in the text;
  • we added a section about nanocomposites and ceramic coatings, and a section about the interactions between bioceramic materials and living bone tissues;
  • we checked for minor issues and removed all the off-topic sentences regarding non-ceramic and non-biomedical arguments.

Below we have reproduced the Reviewers’ comments in bold typeface and given our reply in italics. Where necessary, we have included parts of the revised manuscript in our replies in inverted commas.

Please let us know if the revised paper now satisfies all requirements for publication.

Should you have any questions, please do not hesitate to contact me.

Thank you very much in advance for your attention and courtesy.

Yours Sincerely,

Antonio Boccaccio, Stefania Cantore and Andrea Ballini

REPLY TO REVIEWER 3

Development of ceramic nanocomposite is one of the latest trends for creating materials with outstanding properties. Hence, it is suggested to add a brief paragraph on ceramic nanocomposites developed for biomedical applications. Explain how bioactive molecules are coated on the surface of the ceramic implants to develop functional bioceramics. Also, to reduce the risk of implant related infections or to prevent failed implants, surface of the implants are coated with antimicrobial agents. Cite few important studies.

We sincerely thank the Reviewer for this comment that helped us to improve the quality of the manuscript. We added whole sections for discussing the topics of ceramic nanocomposites, ceramic coatings and bioactive ceramics, as in the section “Advanced bioceramics” (please, see from line 268 of the revised manuscript). Some parts added in the manuscript and addressing the highlighted points are reported in the following.

From “Advanced bioceramics”, (please, see from line 277 to line 282 of the revised manuscript):

“The characteristic trait of nanocomposite materials is their ability to combine properties and functionalities that are out of reach for traditional materials. By incorporating nanoparticles into a ceramic matrix (e.g. by adding organic molecules, carbon nanotubes, graphene, nanoscale ceramics, proteins or even DNA to bioceramics or bioglasses), it is possible to create materials with improved mechanical strength, biocompatibility and osteoconductivity. […]

(please, see from line 334 to line 338 of the revised manuscript):

[…]. Another outstanding development in bioceramic field has been the creation of bioactive coatings. The surfaces of metal or ceramic implants can be coated with ceramic layers, bioactive molecules or antimicrobial agents for preventing the risk of infections and promoting tissue regeneration, wound healing, and osteointegration with the surrounding tissues, thus obtaining an effective functional material.”

Round 2

Reviewer 1 Report

Thank you for revising the manuscript.